# Ethnic Identity as a Mediator of the Relationship between Discrimination and Psychological Well-Being in South—South Migrant Populations

**DOI:** 10.3390/ijerph18052359

**Published:** 2021-02-28

**Authors:** Alfonso Urzúa, Alejandra Caqueo-Urízar, Diego Henríquez, Marcos Domic, Daniel Acevedo, Sebastian Ralph, Gonzalo Reyes, Diego Tang

**Affiliations:** 1Escuela de Psicología, Universidad Católica del Norte, Antofagasta 1240000, Chile; xdiegohenriquez@gmail.com (D.H.); mdomic@ucn.cl (M.D.); 2Instituto de Alta Investigación, Universidad de Tarapacá, Arica 1000000, Chile; acaqueo@academicos.uta.cl; 3Escuela de Psicología y Filosofía, Universidad de Tarapacá, Arica 1000000, Chile; daniieel.ignacio7@gmail.com (D.A.); sebastian.ralph.ponce@gmail.com (S.R.); mr.resille@gmail.com (G.R.); diego.mineria3d@gmail.com (D.T.)

**Keywords:** psychological well-being, discrimination, ethnic identity, migration, racism, well-being

## Abstract

There is abundant evidence about the negative impact of discrimination on well-being, but less research on factors that can reduce this negative effect, mainly focused on North American samples and with incipient development on South–South migration. The objective of this research was to analyze the effect of ethnic identity on the relationship between the experience of racial and ethnic discrimination and psychological well-being in Colombian immigrants living in Chile. A total of 962 immigrants over the age of 18 from three cities in Chile participated. Of these, 50.7% were women. The average age was 35 years (SD = 10.23). Participants were evaluated using Ryff’s Psychological Well-Being Scales, Phinney’s adapted version of the Multigroup Ethnic Identity Scale, and Krieger’s Discrimination Experience Scale. After the analysis of the measurement models, a mediation model was analyzed using structural equations. The results provide evidence that ethnic and racial discrimination have negative effects on psychological well-being, with the effect of racial discrimination being greater. Likewise, ethnic identity has positive effects on psychological well-being and partially and completely mediates the effects of ethnic and racial discrimination on psychological well-being. The full effect of discrimination on psychological well-being, mediated by ethnic identity, is exercised only by racial discrimination and not by ethnic discrimination.

## 1. Introduction

Migration can be understood as the movement of people out of their place of habitual residence for a variety of reasons, either across an international border or within a state, temporarily or permanently [1]. It is estimated that in 2019, 272 million people lived in a country other than their country of birth [2]. By 2050, there will be at least 405 million international migrants [3].

Upon arrival in the host country, migrants try to adapt to their surrounding environment with which they must interact, which can sometimes facilitate integration. Other times it can be difficult, especially when there is discrimination against the migrant population. This is one of the main factors reported to negatively impact migrants’ health and well-being [4,5].

### 1.1. Discrimination and Well-Being

Discrimination can be defined as the different treatment of a group with common characteristics or of people who belong to that group [6] and can be a stressful event that places greater psychological demands on people who suffer it [7,8,9,10,11]. There is abundant evidence about the negative effect that discrimination has on the health, well-being, and quality of life of the people who suffer it [12,13,14,15,16,17,18,19,20,21,22,23,24,25].

There are multiple forms of discrimination, the most frequent among the migrant population being ethnic (ED) and racial discrimination (RD). RD refers to discrimination based on a person’s skin color or physical characteristics, while ED is based on a person’s culture [26]. In the specific case of the migrant population, the research reports a strong impact both on their health [27,28,29,30,31,32,33,34,35] and on their quality of life and well-being [36,37,38,39,40]. 

Studies in the field of well-being are approachable mainly from two perspectives. On the one hand, conceiving well-being from a hedonic perspective (where the main object of study is subjective well-being), and on the other, from a eudaimonic perspective, where psychological well-being is one of the most studied concepts. Psychological well-being is understood as a state of harmony and psychological fullness, where growth and personal development are the main factors of positive functioning [41]. 

Despite the abundant evidence of the negative effect of discrimination on health and well-being, research on factors that can moderate or mediate this relationship is still scarce, especially in South American migrants, constituting an incipient line of research, given its possible impact on the development of preventive programs. In the general population, there are reports on the mediating role of personality traits [42], ethnic affirmation [43], atheist identification [44], or sense of control [45]. In the immigrant population, there are reports of both moderating factors such as group efficacy [46], group membership [47] and identity [48,49], as well as mediating factors such as self-esteem [50], affect [51], employability [52], and group identity [53]. 

### 1.2. Ethnic Identity

There is evidence that ethnic identity is positively related to psychological well-being [16,54,55,56,57] and that it also has a mediating effect on the sense of psychological well-being [58,59,60,61] and moderating role in the relations between perceived RD and psychological distress on Asian immigrants [62]. Ethnic identity is understood as a lasting and fundamental aspect of the self that includes a sense of belonging to an ethnic group and the feelings and attitudes associated with that membership [63], influencing the strategies that a migrant may use to bond with a society [64]. Therefore, well-being can be derived from this bond, since in situations of discrimination, migrants can exacerbate or disguise their ethnic identity to adapt to their new environment [65].

### 1.3. The Current Research

This research is contextualized in south–south migration, that is, from a South American country to another South American country, as is the case of Colombian migration to Chile. The migrant population in the country has increased from 0.81% of the total population in 1992 to 6.55% in February 2019 [66,67]. The immigration of Colombians to Chile is part of the phenomenon of south–south migration, with motivations ranging from the search for job stability to the need to leave vulnerable environments, mainly due to political conflicts in migrants’ countries of origin [68]. The Colombian population in Chile has reported multiple experiences of discrimination, either by the country of origin, linked to drug trafficking, drugs, and sex trade in the case of women, or based on the color of the skin (many migrants are African descendants) [69,70,71]. Previous studies in Chile reported how discrimination has negatively affected both the mental health and the well-being of this population [72,73], in addition to other factors that also affect well-being [74,75,76,77].

The relationship between psychological well-being, identity, and discrimination in Chile has been investigated previously [16], but instead of focusing on ethnic and racial discrimination, such research has examined collective identity and social discrimination. Thus, researchers have found that discrimination has a negative influence and collective identity has a positive influence on psychological well-being, and that strengthening the sense of belonging could create environments and living conditions that are conducive to mental health and healthy lifestyles. In a similar context, Liu and Zhao [53] found that identity (group identity) can have a moderating effect on the relationship between discrimination and well-being.

Considering the above, the objective of this study is to analyze the potential mediating role of ethnic identity on the effects of racial and ethnic discrimination on the psychological well-being of Colombian immigrants living in Chile. In this context, the hypothesis of this research is that ethnic identity is one of the mechanisms that can explain the effect of discrimination, whether racial or ethnic, on psychological well-being, having a mediating role, since ethnic identity has a positive relationship with personal well-being [54], which could even counteract the negative influence of discrimination on the onset of depression [78].

## 2. Materials and Methods

### 2.1. Sample

The sample was composed of 962 Colombian immigrants over 18 years old—484 from Antofagasta (50.3%), 250 from Arica (26%), and 228 from Santiago (23.7%). The participants were mainly surveyed in Caritas Chile, Consulates of Colombia, health centers, and work domiciles. Of the total number of participants, 486 were women (50.7%). The age of participants ranged from 18 to 89 years, with an average of 35.5 years (SD 10.23).

### 2.2. Measures

To identify respondents’ sociodemographic data, questions were asked relating to age, sex, nationality, level of studies, permanence in the country, city of residence, type of residence, level of income, and work situation (economic activity).

### 2.3. Psychological Well-Being

This variable was evaluated through an ad hoc scale proposed by Carol Ryff [79], composed of 29 items, grouped in a model of six dimensions that assess different areas of personal development: self-acceptance, personal growth, life purpose, environmental mastery, autonomy, and positive relations with others. In this research, we used the adaptation created by Díaz et al. [41], a version that has already been used in Chile, reporting adequate psychometric indicators [80,81], including the Colombian population living in Chile [74].

### 2.4. Perceived Discrimination

The Experience of Discrimination (EOD) scale proposed by Krieger et al. [82], was used to evaluate discrimination. The original scale is composed of 10 items that assess the perception of discrimination on various grounds. For the purpose of this research, participants were asked about their experiences of discrimination due to their skin color, and independently, in another scale, due to their ethnic origin (being Colombian). 

### 2.5. Ethnic Identity

The ethnic identity variable was evaluated using the “Multigroup Ethnic Identity Scale” (MES) developed by Phinney [83] in its adaptation for Spanish-speaking populations in Ibero-American countries carried out by Smith [84] in Costa Rica. The model proposed by Smith considers 12 items grouped into two highly related components of ethnic identity in all the groups studied: ethnic affirmation and identification and ethnic exploration. This instrument reports good psychometric properties and has been used previously with Colombian migrants in Chile [64].

### 2.6. Procedures

The project was reviewed and approved by an accredited institutional scientific ethics committee (Comité de Ética Científica de la Universidad Católica del Norte). As it is impossible to exactly determine the size of the population universe, and therefore the sampling frame of the population of Colombian immigrants over 18 years of age in the relevant cities, we used non-probabilistic sampling, combining the so-called snowball sampling proposed by Goodman [85] with the sampling conducted by the interviewees. Once the seed individuals who would initiate the chains (waves) were contacted, they referenced new participants, who were contacted and invited to voluntarily participate in the research. The objective of the research was explained to participants, and they then signed informed consent forms. The interviewers were final-year psychology students, master’s students, or Colombian migrants, all of whom were trained in the use of the instruments. Since both interviewers and surveyors are native speakers of Spanish, the use of translators was not necessary. The average duration of the survey was approximately 90 min. Once the questionnaires were collected, the data were entered into a database built in SPSS [86]. 

### 2.7. Statistical Analysis

First, the measurement models were tested and adjusted through confirmatory factor analysis of each of the variables included in the hypothesized models. Once the measurement models were estimated, a mediation model was tested with two independent variables (ED and RD), a mediating variable (ethnic identity), and the dimensions of psychological well-being as dependent variables (self-acceptance, positive relationships, autonomy, environmental mastery, and personal growth). The indirect effects of the mediation model were estimated following the recommendations of Stride et al. [87]. The models’ goodness-of-fit were estimated using chi-square values (χ^2^), the approximation mean square error (RMSEA), the comparative adjustment index (CFI), and the Tucker–Lewis index (TLI). For all analyses, the robust maximum-likelihood ratio (MLR) method was used, which is robust to the assumption of multivariate normality [88]. It is important to point out that in the models where ED was included along with RD, the errors of the items with similar wording were correlated, equivalent between both variables (e.g., You have felt discriminated against in your work: (ED) “...for being Colombian” and (RD) “...for your skin color”). The statistical packages used were SPSS [86], Jamovi v. 0.9 (Jamovi project, Sydney, Australia), and MPlus v. 8.2 (Muthen & Muthen, Los Angeles, CA, USA).

## 3. Results

### 3.1. Measurement Models

Table 1 shows the goodness-of-fit indices of the measurement models that were analyzed. The measurement models of the Ryff scale and the ethnic identity scale presented poor levels of fit, with indexes lower than the standards recommended by the literature (RMSEA < 0.08; CFI > 0.95; TLI > 0.95). For this reason, it was decided to iteratively refine the initial models, reducing the number of items in each scale. 

Specifically, the Ryff scale measurement model presented some Heywood cases (standardized correlations over 1) between the Life Purpose and Environment Domain variables, and between Life Purpose and Self-Acceptance. Because small factor loads can be the cause of Heywood cases [89], we decided to test alternative models by iteratively removing items with non-significant factor saturations. The items with non-significant factorial saturations were “I often feel lonely because I have few close friends with whom to share my concerns” (Positive Relationships), “I find it difficult to direct my life towards a path that gives me satisfaction”, “I am confident in my opinions even if they are contrary to what the majority thinks” (Autonomy), “I tend to worry about what other people think of me” (Purpose in Life) and “I think that over the years I have not improved much as a person” (Personal Growth). The fit rates of the alternative measurement model improved but fell short of the recommended criteria (CFI = 0.87; TLI = 0.85), and the Heywood cases continued. For this reason, we decided to test a model without considering the Purposes in Life dimension because it was the dimension that had the most associated Heywood cases. The model finally used for the analyses is composed of 13 items grouped into 5 factors: self-acceptance (4 items) and positive relationships, environmental mastery and personal growth, each with 3 items. The Cronbach’s alpha coefficient of each one of the dimensions of the scales was estimated: self-acceptance = 0.79; positive relationships = 0.70; autonomy = 0.77; environmental mastery = 0.62; and personal growth = 0.82. This last alternative measurement model showed no Heywood cases and good, but not excellent, adjustment rates (see Table 1); so, it is necessary that in future research the psychometric properties of this version of the Ryff scale be further analyzed. 

In the case of the ethnic identity scale, although it did not present any Heywood cases, it did show items with non-significant factor loads (“Sometimes he/she prefers to hide that he/she is Colombian”) and/or small ones (“Being Colombian does not define what you really are”). Despite having iteratively removed items with small, non-significant factor loads, the model still presented poor adjustment rates (RMSEA > 0.8; CFI < 0.9). Under these conditions, we continue with an analysis on the modification indexes of the last alternative measurement model. Three pairs of items were identified that could be sharing excessive variance of the content they seek to reflect. These pairs of items are as follows: “You are happy to be Colombian” with “You feel very good about your cultural tradition”; “Being Colombian defines very well what you are” with “You could say you are a typical Colombian”; and “You are aware of your Colombian roots” with “You feel very attached to your country and culture”. Therefore, it was decided to try alternative models, iteratively removing the items with lower factorial saturations than their opponent (“You are happy to be Colombian”, “Being Colombian defines very well what you are”, and “You are aware of your Colombian roots”). The model finally used, and which presented the best fit, corresponds to a version of 5 items grouped into a single general factor. This scale had a Cronbach’s alpha of 0.85. This alternative model turned out to have adequate goodness of fit indices (see Table 1). However, it is important to note that future research should analyze the psychometric properties of this version of ethnic identity in greater depth.

The models for measuring ED and RD presented good indices of adjustment and therefore are good representations of the relationships observed and did not need any modification. The Ethnic Discrimination Scale presented a reliability of 0.92, and finally the Racial Discrimination Scale presented a Cronbach’s alpha coefficient of 0.84.

### 3.2. Structural Equation Models

Once the models of measurement were adjusted, we proceeded to examine the effects of ED and RD on the dimensions of psychological well-being (Figure 1).

It was observed that ED has small (*b* > 0.10) inverse effects on self-acceptance (*b* = −0.22; *p* = 0.001) and personal growth (*b* = −0.15; *p* = 0.025), and a moderate (*b* > 0.30) inverse effect on environmental control (*b* = −0.34; *p* < 0.001) [90]. ED did not present statistically significant effects on positive relations and autonomy. On the other hand, RD presented only a small inverse effect on positive relations (*b* = −0.17; *p* = 0.025). RD did not present statistically significant effects on the rest of the dimensions of psychological well-being (self-acceptance, autonomy, control of the environment, and personal growth). The structural model presented goodness-of-fit close to the criteria accepted in the literature (RMSEA = 0.047; CFI = 0.905; TLI = 0.895).

Once the relationship between both types of discrimination and psychological well-being was examined, the model of simple mediation was evaluated. In this model, the ethnic identity on migrants were included as mediator of the inverse effect that ED and RD would have on dimensions of psychological well-being (self-acceptance, positive relations, autonomy, environmental mastery, and personal growth, see Figure 2). Figure 2 shows that ED has small positive effects on autonomy (*b* = 0.13; *p* = 0.40) but negative effects on ethnic identity (*b* = −0.25; *p* < 0.001), self-acceptance (*b* = −0.13, *p* = 0.040), and environmental control (*b* = −0.27; *p* < 0.001). As for RD, only a small and inverse effect on positive relations can be observed (*b* = −0.18; *p* = 0.020).

The direct, indirect, and total effects of the estimated mediation model can be seen in Table 2. As can be seen, ethnic identity demonstrates a total mediation of the effects that ED has on positive relationships and personal growth. Similarly, ethnic identity demonstrates a partial mediation of the effects that ED has on self-acceptance, autonomy, and control of the environment. No statistically significant indirect effects of ethnic identity on the relationship between RD and the dimensions of psychological well-being were found (see Table 2). The hypothesized mediation model was adequately adjusted to the data (RMSEA = 0.044; CFI = 0.909; TLI = 0.900); therefore, it was a good representation of the observed relationships.

According to the results of the mediation model, ethnic identity would only present mediation effects on the relationship between ED and the dimensions of psychological well-being. In addition, it is important to note that although the effects observed in the models are small, these effects could have important consequences if they are recurrent events over time [91].

## 4. Discussion

The objective of this study was to analyze the relationship between the experience of ethnic and racial discrimination and the perception of psychological well-being in relation to ethnic identity in Colombian immigrants currently living in Chile.

The central hypothesis in this study was that ethnic identity has a mediating effect on the relationship between ethnic and racial discrimination and psychological well-being.

The results indicate that both types of discrimination do indeed have inverse effects on psychological well-being. These results are like those of García et al. [16], who showed that discrimination, specifically social discrimination, negatively influenced psychological well-being. Further, ED and RD both have small effects on the dimensions of psychological well-being. Regarding one’s perception of their dominance over their environment, greater perceptions of prejudice are accompanied by lower levels of internal well-being. It is possible that such relations do not stop people from “enjoying life” but that such experiences place tensions on this reality. This is a tentative explanation for why the levels of overall well-being in the sample do not differ from those measured in other non-minority populations. The effects in the dimension of the environment are moderate; thus, discrimination may make migrants feel that they have a lack of control of their lives [92].

The literature shows that ethnic identity functions as a protective factor; that is, one’s feeling of belonging to an inner group diminishes the influence of the variables that exert negative effects. This has already been pointed out by Hughes et al. [93], who showed that ethnic identity, and specifically ethnic affirmation, mediates the effects of cultural socialization on academic results. Likewise, ethnic identity exerts a simple partial mediation on the relationship between ethnic discrimination and the dimensions of self-acceptance, positive relations, control of the environment, and personal growth; a simple complete mediation between ED and personal growth; and a simple partial mediation between RD and all dimensions of psychological well-being. In other words, one’s sense of belonging to an ethnic and/or phenotypic inner group could mitigate the influence of discrimination in the different domains that comprise one’s psychological well-being.

This mediation of ethnic identity has been identified by other authors who have found that group identification is generated as a defense mechanism against discrimination. Group identification strengthens self-esteem, which visibly influences psychological well-being [94]. This mediating role of ethnic identity on present discrimination has also been shown to partially measure the effects on depression and anxiety [95]. On the other hand, Huq et al. [96] mention that in context of stress resulting from ED, young people may develop an ethnic identity to protect their psychological well-being. Thus, ethnic identity may be a protective factor for those who have previously developed it but not necessarily in young people and children.

On the other hand, when ED and RD are presented as a whole with ethnic identity as a mediator, the inverse indirect effects exerted by ED on the dimensions of psychological well-being are no longer statistically significant, and only the small direct inverse effects exerted on the dimension of positive relationships remain. In this sense, ethnic identity can function as a protective measure; it has been found that when members of stigmatized groups recognize that they are victims of discrimination, they may increase their identification with their ethnic group as part of their coping strategies, countering the negative impact that ED has on individual self-worth and self-esteem [50].

Bearing this in mind, it can be said that since both types of discrimination are presented as a whole with ethnic identity as a mediator, the negative effects of RD on psychological well-being are effectively diminished. On the other hand, when ED is presented together with RD, it does not have a significant effect on psychological well-being; this may be because people pay more attention to what is visible, such as migrants’ skin color or predominant features, than to cultural aspects. These results point to a direction contrary to that reported by Yoo and Lee [97], who found that ethnic identity may exacerbate the association between RD and situational well-being for Asian Americans, especially for U.S.-born Asian Americans. This could give light to a future line of research, as sociocultural context may also affect the buffer effect that has been identified. Many studies consider only one type of discrimination either racial [97,98] or ethnic [99].

Another interesting result was the high correlation found between ED and RD. From the theoretical perspective of the intersectionality between multiple identities [100], in this case, migrants who present a dual identity as migrants and also as Black people (or indigenous people) could be more exposed to discrimination and consequently undermine their psychological well-being [101]. For this reason, further analyses addressing this issue should be conducted in the future in order to expand the scope of the results presented in this study.

One of the strengths of this study was that both racial and ethnic discrimination were considered, which can provide greater insights on the effect of ethnic identity on psychological well-being, allowing to describe and conclude that ethnic identity mediates the effects of both racial and ethnic discrimination. Secondly, another strength that can be mentioned is the possibility of addressing research questions regarding Latin American migration, especially on the context of south–south migration, which are lacking, and can open potential lines of research enriching what is known about ethnic identity, well-being, and discrimination. Thirdly, several studies on ethnic identity, discrimination, and well-being have been conducted on adolescents in the Latino population [99,102]. However, studying a different cohort of participants is crucial because there might be interesting differences between adolescents and adults’ experiences. Additionally, analyzing and determining the impact of discrimination on specifics components of psychological well-being, such as self-acceptance, positive relations, control of the environment, and personal growth, not only could be new but also might provide a body of knowledge available that could serve to elaborate potential public policies.

In terms of the limitations of this study, it should be noted that by considering only the Colombian population, the generalization of these results to other immigrant communities should be done with caution. Another limitation of this study is related to the deficit of research on immigrants in Chile and in other Latin American countries, where the panorama is comparable. This lack of similar work hinders the ability to contrast the results and understand situations with immigrants in comparable contexts.

It is worth mentioning that it is important to continue studying immigration and deepening the understanding of this complex psychosocial and geopolitical phenomenon. It is also urgent that we review our public policies (immigration laws, education, health, work, etc.) and develop updated and reliable information on migrants at the national level in order to be able to promote future analyses, with the aim of covering new trends that are developing, e.g., the problems associated with migration, and to permanently evaluate occurrences in this area. On the other hand, given the multidimensional nature of the “ethnic identity” construct and the fact that there is no absolute consensus about the dimensions that compose it, we believe that it is also necessary to disaggregate the effect that each of its components may have on the relationship studied or on well-being. Moreover, in this study, we focus on the fact of feeling discriminated against by skin color, regardless of what it is, since there is not only discrimination against Black people, but also against mulattoes, mestizos, and those of indigenous descent. We believe that it is important in future research to expand the sample size in order to stratify the analysis by racial self-identification. Finally, considering that previous research has shown that the developmental trajectories of racial/ethnic identities could differ according to the age and developmental period of the participants’ lives, it would be interesting to evaluate in future research the possible moderating effect of age on the mediating effect of ethnic identity.

This research contributes to the development of a better understanding of the Colombian migration phenomenon in Chile. Further, it lays the groundwork for future research and interventions in the area, particularly for government programs based on the mediating role of ethnic identity in the migrant population, which offers comprehensive interventions, considering the complexities in the variables studied.

## 5. Conclusions

Finally, by way of conclusion and in response to the objective, the experience of ethnic and racial discrimination effectively could influence psychological well-being, and ethnic identity could mediate this effect; that is, it diminishes the effects of racial and ethnic discrimination on the perception of psychological well-being in Colombian immigrants currently living in Chile.

## Figures and Tables

**Figure 1 ijerph-18-02359-f001:**
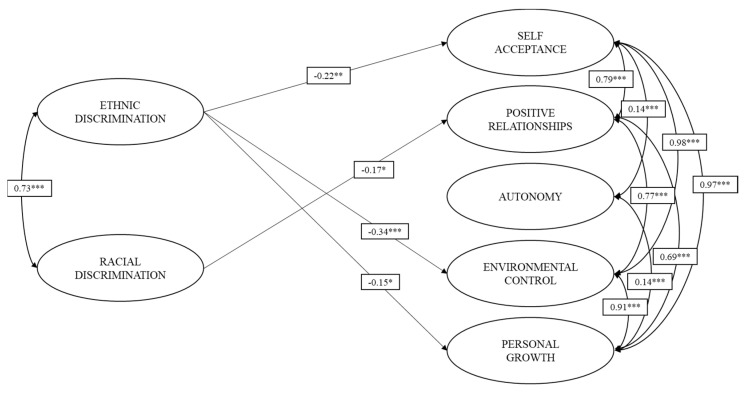
Relationship between ethnic and racial discrimination and the domains of psychological well-being. Only the statistically significant effects of the model are plotted. * *p* < 0.05. ** *p* < 0.01. *** *p* < 0.001.

**Figure 2 ijerph-18-02359-f002:**
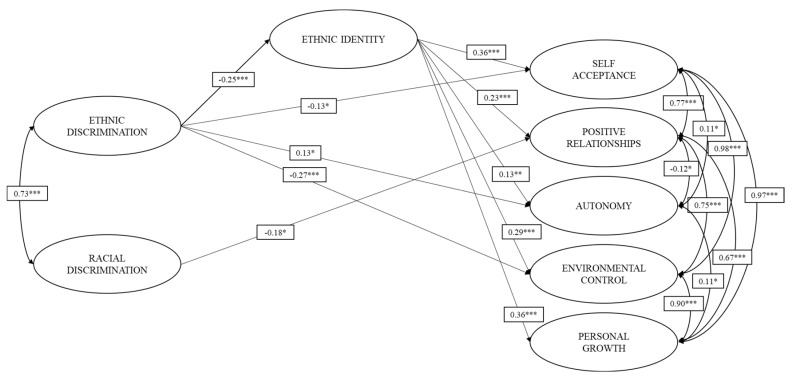
Mediating effect of ethnic identity on the relationship between ethnic and racial discrimination and well-being. Only the statistically significant effects of the model are plotted. * *p* < 0.05. ** *p* < 0.01. *** *p* < 0.001.

**Table 1 ijerph-18-02359-t001:** Overall adjustment indicators of the measurement models.

Models	Parameters	χ^2^	DF	*p*	CFI	TLI	RMSEA	RMSEA IC 90%	
								Below	Superior
Ryff ^a^	102	2860.721	362	0.000	0.727	0.694	0.085	0.082	0.088
Ryff ^b^	61	481.444	109	0.000	0.914	0.893	0.060	0.054	0.065
IE ^c^	30	567.276	35	0.000	0.806	0.751	0.126	0.117	0.135
IE ^d^	15	30.400	5	0.000	0.973	0.945	0.073	0.049	0.099
DE	27	197.898	27	0.000	0.921	0.894	0.081	0.071	0.092
DR	27	124.126	27	0.000	0.945	0.927	0.063	0.052	0.075

^a^ Ryff’s original model; ^b^ adjusted model; ^c^ original model; ^d^ adjusted model. CFI, comparative adjustment index; TLI, Tucker–Lewis index; RMSEA, root mean square error of approximation.

**Table 2 ijerph-18-02359-t002:** Standardized indirect and total effects of the mediation model.

**Effects**	**Direct**	**Indirect**	**Total**
ED→IE→SA	−0.128 *	−0.091 ***	−0.219 **
ED→IE→PR	−0.076	−0.056 **	−0.132
ED→IE→AU	0.132 *	−0.032 *	0.100
ED→IE→EM	−0.272 ***	−0.072 **	−0.334 ***
ED→IE→PG	−0.063	−0.090 **	−0.153 *
RD→IE→SA	−0.015	0.013	−0.002
RD→IE→PR	−0.182*	0.008	−0.174 *
RD→IE→AU	−0.052	0.005	−0.048
RD→IE→EM	0.049	0.011	0.059
RD→IE→PG	−0.031	0.013	−0.018
**Effects**	**Direct**	**Indirect**	**Total**
ED→IE→SA	−0.128	−0.091 *	−0.219 *
ED→IE→PR	−0.076	−0.056 *	−0.132
ED→IE→AU	0.132 *	−0.032 *	0.100
ED→IE→EM	−0.272 *	−0.072 *	−0.334 *
ED→IE→PG	−0.063	−0.090 *	−0.156 *
RD→IE→SA	−0.015	0.013	−0.002
RD→IE→PR	−0.182 *	0.008	−0.174 *
RD→IE→AU	−0.052	0.005	−0.048
RD→IE→EM	0.049	0.011	0.059
RD→IE→PG	−0.031	0.013	−0.018

Note: RD = racial discrimination; ED = ethnic discrimination; IE = ethnic identity; SA = self-acceptance; PR = positive relationships; AU = autonomy; EM = environmental mastery; PG = personal growth. * *p* < 0.05. ** *p* < 0.01. *** *p* < 0.001.

## Data Availability

The data presented in this study are available on request from the corresponding author. The data are not publicly available because the project has state funding and will only be released once the project is finished.

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
