# Peer review of "Ethnic Identity as a Mediator of the Relationship between Discrimination and Psychological Well-Being in South—South Migrant Populations"

_ijerph, 2021, doi:10.3390/ijerph18052359_

Round 1

Reviewer 1 Report

General:

  1. The concept of ‘identity’ is problematic. It is not adequately defined. We need a longer discussion and better definition of what the authors mean by 'identity'. Also, maybe it is better to avoid writing ‘identity’ and to write, instead, identification with and participation in the life of the local ethnic community of colombians, or something like that? Also, on p.6, ‘ethnic identity’ really refers to ethnic consciousness. Maybe use ethnic consciousness or ethnic identification, or ethnic affirmation, instead of 'identity' as you do in other passages?
  2. The existence of two types of colombians in chile, black and non-black needs to be discussed more fully.

Line: 46

Unclear why is disability included in the passage – it diverts attention away form the focus of the research: Discrimination can be defined as the different treatment of a group with common characteristics or of people who belong to that group [6] and can be a stressful event that 47 requires greater psychological demands from people who suffer from disabilities [7-11].

LINE: 90

either by the 90 country of origin, linked to drug trafficking, drugs, and sex trade in the case of women, or 91 based on? by the color of the skin (many migrants are African descenDANts) [70-72].

LINE: 97

ethnic are you not referring to racial identitiy here, for some particular reason? identity and 97 ethnic-racial discrimination, such research has examined collective identity and social

LINE: 122

To identify respondents’ socio-demographic data, questions were asked relatedRELATING 122 to age, sex, nationality, level of studies, p

LINE: 144

The project was reviewed and approved by an accredited institutional scientific 144 ethics committee WHAT IS ITS NAME?

LINE: 360

The words comparative and comparable could be better in the paragraph: Another limitation of this study is related to the deficit of research on immigrants in 360 Chile and in other Latin American countries, where the panorama is similar. This lack of 361 similar work hinders the ability to contrast the results and understand situations with 362 immigrants in similar contexts.

Line: 199

Do you really want to use "it"?s ('Sometimes it prefers to hide 198 that it is Colombian') and/or small ones ('Being Colombian does not define what you 199 really are').

Author Response

  1. The concept of ‘identity’ is problematic. It is not adequately defined. We need a longer discussion and better definition of what the authors mean by 'identity'. Also, maybe it is better to avoid writing ‘identity’ and to write, instead, identification with and participation in the life of the local ethnic community of colombians, or something like that? Also, on p.6, ‘ethnic identity’ really refers to ethnic consciousness. Maybe use ethnic consciousness or ethnic identification, or ethnic affirmation, instead of 'identity' as you do in other passages?

Answer: We understand your concern, however, we have preferred to maintain the global concept of ethnic identity, specifically referring to the sense of belonging to an ethnic group, which, being multidimensional, incorporates various elements such as a sense of belonging, ethnic awareness, preference for the group ethnicity, interest, knowledge and participation in activities associated with the group. This according to the definition of Salas, based on the proposal of Phinney & Rosenthal (1992). In this way, it is congruent with the instrument used and with the previous analysis of the factorial structure, which, for the purposes of the analysis carried out in this work, allowed to have a univariate structure that incorporates various components of the construct. This point is detailed later in the instruments section. Since we find the author's observation very interesting, we have added a paragraph in the discussion about this.

Given the multidimensional nature of the "ethnic identity" construct and the fact that there is no absolute consensus about the dimensions that compose it, we believe that it is also necessary to disaggregate the effect that each of its components may have on the relationship studied or on well-being.

2. The existence of two types of colombians in chile, black and non-black needs to be discussed more fully.

Answer: We are aware of this, however, in this study we focus on the fact of feeling discriminated against by skin color, regardless of what it was, since there is not only discrimination against black people, but also against mulattoes, mestizos and of indigenous descent. we have added a paragraph about this in the discussion, thank you very much !!!

In this study, we focus on the fact of feeling discriminated against by skin color, regardless of what it is, since there is not only discrimination against black people, but also against mulattoes, mestizos and those of indigenous descent. We believe that it is important in future research to expand the sample size in order to stratify the analysis by racial self-identification

3. Line: 46 Unclear why is disability included in the passage – it diverts attention away form the focus of the research: Discrimination can be defined as the different treatment of a group with common characteristics or of people who belong to that group [6] and can be a stressful event that 47 requires greater psychological demands from people who suffer from disabilities [7-11].

Answer: It was a mistake; the wording was corrected and the word disability removed

Discrimination can be defined as the different treatment of a group with common characteristics or of people who belong to that group [6] and can be a stressful event that requires greater psychological demands from people who suffer it [7-11].”

4. LINE: 90 either by the 90 country of origin, linked to drug trafficking, drugs, and sex trade in the case of women, or 91 based on? by the color of the skin (many migrants are African descenDANts) [70-72].

Answer: thanks for the correction, these were made in the text

5. LINE: 97 ethnic are you not referring to racial identitiy here, for some particular reason? identity and 97 ethnic-racial discrimination, such research has examined collective identity and social

Answer: It was a mistake; the wording was corrected

“The relationship between psychological well-being, identity, and discrimination in Chile has been investigated previously [16], but instead of focusing on ethnic and racial discrimination…”

6. LINE: 122 To identify respondents’ socio-demographic data, questions were asked relatedRELATING 122 to age, sex, nationality, level of studies, p

Answer: thanks for the correction, these were made in the text

7. LINE: 144 The project was reviewed and approved by an accredited institutional scientific 144 ethics committee WHAT IS ITS NAME?

Answer: We have added the requested name in the text

The project was reviewed and approved by an accredited institutional scientific ethics committee (Comité de Ética Científica de la Universidad Católica del Norte).

8. LINE: 360 The words comparative and comparable could be better in the paragraph: Another limitation of this study is related to the deficit of research on immigrants in Chile and in other Latin American countries, where the panorama is similar. This lack of similar work hinders the ability to contrast the results and understand situations with immigrants in similar contexts.

Answer: We have incorporated the reviewer's wording suggestion

Another limitation of this study is related to the deficit of research on immigrants in Chile and in other Latin American countries, where the panorama is comparable. This lack of similar work hinders the ability to contrast the results and understand situations with immigrants in comparable contexts.

9. Line: 199 Do you really want to use "it"?s ('Sometimes it prefers to hide 198 that it is Colombian') and/or small ones ('Being Colombian does not define what you 199 really are').

Answer: Thank you very much for this correction!!!!!!!. We have changed the wording

In the case of the ethnic identity scale, although it did not present any heywood cases, it did show items with non-significant factor loads ('Sometimes he/she prefers to hide that he/she is Colombian') and/or small ones ('Being Colombian does not define what you really are').

Reviewer 2 Report

The present research examined the relationship between ethnic and racial discrimination on psychological well-being among south-south migrants, as well as whether these relationships were mediated by ethnic identity. This research is very important because it extends current research on these topics into new populations and demographics to better highlight the role of discrimination in shaping the psychological well-being of Colombian immigrants living in Chile.

Abstract + Introduction

In the abstract the authors start by saying there is little research examining factors that can reduce the negative effects of discrimination on well-being. However, among American samples there is ample research over many decades examining this exact question. One of the unique elements this manuscript is that it extends this well-established body of literature to an understudied population: Colombian immigrants living in Chile. As a result, I recommend contextualizing the abstract with the existing body of literature in other nations, then clarifying the unique contribution of the current manuscript.

I appreciate how the authors begin to contextualize the migration history and social climate faced by the focal population in this study. However, I would like more contextual information on the frontend of the introduction about the target population. For example, the authors do not begin to address the specific contextual factors related to the south-south migration population until line 83. Moving this section earlier in the introduction will help emphasize that the authors are testing a well-studied mechanism within a specific/understudied population. Without this clarification, readers may feel as if they are reading about a series of relationships that have already been well-established.

In the introduction of the manuscript (line 46), the authors provide the definition of discrimination they will be using for the present research. Towards the end of the definition they describe the psychological toll of discrimination as being “a stressful event that requires greater demand from people who suffer from disabilities”. The authors may want to change the definition of discrimination they intend to use for the study given that being a migrant/immigrant/POC is not a disability.

Introduction: Moderation vs Mediation

The introduction was somewhat disorganized, which made it hard to follow the overarching direction of the manuscript leading up to the final research question and hypotheses. Specifically, the authors should provide better organization to highlight the distinction between ethnic identity as a mediator vs. moderator. For instance, on line 64 the authors state, “…research on factors that can moderate or mediation this relationship is still scarce…” This phrasing leads readers to expect that the paper will examine both mediation and moderation, yet the paper does not examine both methods.

Additionally, the authors list numerous studies that examine their key variables through mediation and moderation, so they should reconsider describing this research as “scarce”.

Overall,  a more consistent narrative throughout the manuscript is needed to justify the use of mediation analysis as the ideal analytical framework for this paper. The authors should also consider whether moderation analyses would be more appropriate to address the research questions.

To better organize the paper, the authors should include headings and subheadings to help guide the readers through the paper. Currently, some paragraphs appear to be out of place and do not have a natural flow, which disorients the reader. For instance, it would help to have a section specifically dedicated to the direct effect of discrimination on psychological well-being, and then incorporate a separate section explicitly dedicated to how ethnic identity may mediate that relationship.

Methods/Results/Discussion

One major issue is whether the operationalization of racial discrimination is truly measuring racial discrimination, or if it is more accurately measuring skin-tone discrimination (i.e., colorism). The authors noted that they used the Experience of Discrimination scale proposed by Krieger et al. to evaluate discrimination. Looking at that paper, the prompt explicitly says, “have you ever experienced discrimination…because of your race, ethnicity, or color.” However, in the present manuscript, the authors write that they asked participants whether they were discriminated against “due to their skin color” or “their ethnic origin (being Colombian).” Please clarify how racial discrimination and ethnic discrimination were independently operationalized.

If participants were only asked about skin-tone based discrimination, the authors may need to go back through the introduction and adapt the conceptual framework to incorporate colorism, which specifically encapsulates skin-tone based discrimination.

The authors should also provide more information about the specific scales used in the method section. For example, a brief scale description, sample item for each subscale, the number of items in each subscale, and reliability statistics would all be helpful. The authors mention a few of the items as they begin to adjust their measurement models, but systematically presenting items in the method section would be easier for reader to follow.

It would would be helpful to provide more methodological details about the setup of the interviews/data collection. For instance, what was the race/ethnicity of the interviewers? What language were the interviews conducted in? Was a translator needed?

In the results section, it would be helpful to see the exact significance levels reported in either the main text or in the tables/figures.

In the results section on lines 255-257, there is an error in the reported results for the outcome positive relations because ethnic identity did not fully mediate the relationship between ethnic discrimination and positive relations since there was not a statistically significant direct effect between these variables.

Similar for the outcome variable for autonomy, ethnic discrimination did not have a significant direct effect on this outcome variable, so ethnic identity cannot partially mediate a direct effect that did not occur.

Given the cross-sectional nature of the data, the authors should avoid using causal language to describe the mediational results. In the discussion section, the authors describe their findings as occurring because ethnic discrimination caused people to feel more connected to their ethnic group, which in turn, lead to greater psychological outcomes.

One question for the authors to further consider is whether the mediating processes they examined are dependent on the age of the participants. For example, prior research has shown that the developmental trajectories for racial/ethnic identities differ based on age and the developmental period of the participants’ life. As a result, if older adults are more stable in their racial/ethnic identity, they may be less likely to be susceptible to the mediating mechanisms described in this paper.

Small details:

Ethnic Discrimination vs. Racial Discrimination

In the introduction, the authors note that the abbreviation of ED and RD would be used to represent ethnic discrimination and racial discrimination, respectively. However, throughout the course of the manuscript the authors are inconsistent regarding their usage of ED, RD or the full phrases.

Likewise, there are other ways ED and RD are referenced during the paper that aren’t consistent. For example, in the description for Figure 1, but not figure 2, the authors say the relationship between “ethnic/racial discrimination”, whereas, in the paper the authors often refer to them as separate constructs (i.e., racial and ethnic discrimination). Given that these two variables are not collapsed into one, the authors should consistently separate these words out throughout the paper and figure descriptions.

Another instance where the authors deviate from their usual pattern of describing ED and RD is on line 241 where they phrase it as “ethnic-racial discrimination”. In addition, the authors suddenly begin to use the ED and RD abbreviations again on line 243 despite not using it throughout the majority of the paper.

The authors often go back and forth between saying “ethnic and racial discrimination” and “racial and ethnic discrimination”

Line 73: change “on color people” to “on people of color”

Line 198: Did the authors mean to write, “Sometimes ‘it’ prefers to hide that ‘it’ is Columbian”? Or should the “it” be replaced with “I” or “you”?

On line 259/260, I believe the authors meant to write that ethnic identity did not mediate the relationship between “racial discrimination and psychological well-being”, and not “ethnic discrimination”.

There are several paragraphs that are only 2 or 3 sentences long and they impacted the overall flow of particular sections. I encourage the authors to find a better way to integrate these brief paragraphs for a smoother transition.

I recommend reviewing the paper for sentence structures, run-on sentences, oxford commas, and grammatical errors. In particular, there are a few grammatical errors in the paragraph starting on line 349.

Author Response

  1. In the abstract the authors start by saying there is little research examining factors that can reduce the negative effects of discrimination on well-being. However, among American samples there is ample research over many decades examining this exact question. One of the unique elements this manuscript is that it extends this well-established body of literature to an understudied population: Colombian immigrants living in Chile. As a result, I recommend contextualizing the abstract with the existing body of literature in other nations, then clarifying the unique contribution of the current manuscript.

Answer: We agree with the reviewer's suggestion and have drafted the following change to the text:

There is abundant evidence about the negative impact of discrimination on well-being, but less research on factors that can reduce this negative effect, mainly focused on North American samples and with incipient development on South-South migration.

  1. I appreciate how the authors begin to contextualize the migration history and social climate faced by the focal population in this study. However, I would like more contextual information on the frontend of the introduction about the target population. For example, the authors do not begin to address the specific contextual factors related to the south-south migration population until line 83. Moving this section earlier in the introduction will help emphasize that the authors are testing a well-studied mechanism within a specific/understudied population. Without this clarification, readers may feel as if they are reading about a series of relationships that have already been well-established

Answer: We have tried to make the changes suggested by the reviewer, however, we believe it does not favour the reader's understanding, so we have not changed the data of the present research at the beginning.  We have rewritten the introduction, incorporating a more logical order and headings, as suggested by the reviewer.

  1. In the introduction of the manuscript (line 46), the authors provide the definition of discrimination they will be using for the present research. Towards the end of the definition they describe the psychological toll of discrimination as being “a stressful event that requires greater demand from people who suffer from disabilities”. The authors may want to change the definition of discrimination they intend to use for the study given that being a migrant/immigrant/POC is not a disability.

Answer: We are very sorry for this editing error. We have corrected it in the text:

Discrimination can be defined as the different treatment of a group with common characteristics or of people who belong to that group [6] and can be a stressful event that requires greater psychological demands from people who suffer it [7-11].

  1. The introduction was somewhat disorganized, which made it hard to follow the overarching direction of the manuscript leading up to the final research question and hypotheses. Specifically, the authors should provide better organization to highlight the distinction between ethnic identity as a mediator vs. moderator. For instance, on line 64 the authors state, “…research on factors that can moderate or mediation this relationship is still scarce…” This phrasing leads readers to expect that the paper will examine both mediation and moderation, yet the paper does not examine both methods.

 Additionally, the authors list numerous studies that examine their key variables through mediation and moderation, so they should reconsider describing this research as “scarce”.

 Overall,  a more consistent narrative throughout the manuscript is needed to justify the use of mediation analysis as the ideal analytical framework for this paper. The authors should also consider whether moderation analyses would be more appropriate to address the research questions.

 To better organize the paper, the authors should include headings and subheadings to help guide the readers through the paper. Currently, some paragraphs appear to be out of place and do not have a natural flow, which disorients the reader. For instance, it would help to have a section specifically dedicated to the direct effect of discrimination on psychological well-being, and then incorporate a separate section explicitly dedicated to how ethnic identity may mediate that relationship.

Answer: We have reordered the introduction as suggested by the reviewer, and we included headings.   At the end of the paragraph on the present study, we have explained that we are interested in assessing the mediation hypothesis, as we believe it is able to explain part of the effect of discrimination on welfare, rather than the change in the effect, which would be a moderation analysis.

  1. One major issue is whether the operationalization of racial discrimination is truly measuring racial discrimination, or if it is more accurately measuring skin-tone discrimination (i.e., colorism). The authors noted that they used the Experience of Discrimination scale proposed by Krieger et al. to evaluate discrimination. Looking at that paper, the prompt explicitly says, “have you ever experienced discrimination…because of your race, ethnicity, or color.” However, in the present manuscript, the authors write that they asked participants whether they were discriminated against “due to their skin color” or “their ethnic origin (being Colombian).” Please clarify how racial discrimination and ethnic discrimination were independently operationalized.  If participants were only asked about skin-tone based discrimination, the authors may need to go back through the introduction and adapt the conceptual framework to incorporate colorism, which specifically encapsulates skin-tone based discrimination.

Answer: As mentioned in the section on instruments, the research has considered the adaptation of Krieger's scale, which asks about discrimination on various grounds, specifying it in two different instruments. On the one hand, racial discrimination (defined in this manuscript as that linked to skin colour) and, on the other, ethnic discrimination.

Participants were asked about their experiences of discrimination due to their skin color, and independently, in another scale, due to their ethnic origin (being Colombian).

Colourism is a construct that is not widely disseminated, which is interesting, but is still considered to be a sub-classification of racial discrimination, which remains the most widely used construct in the literature.

  1.  The authors should also provide more information about the specific scales used in the method section. For example, a brief scale description, sample item for each subscale, the number of items in each subscale, and reliability statistics would all be helpful. The authors mention a few of the items as they begin to adjust their measurement models, but systematically presenting items in the method section would be easier for reader to follow.

Answer:  The reviewer's suggestions are taken on board and the instruments are better detailed both in the instruments section and in the presentation of the measurement models.

  1. It would be helpful to provide more methodological details about the setup of the interviews/data collection. For instance, what was the race/ethnicity of the interviewers? What language were the interviews conducted in? Was a translator needed?

Answer:  The reviewer's suggestions are considered and the procedures are better detailed.

The interviewers were final year psychology students, master's students, or Colombian migrants, all of whom were trained in the use of the instruments.  Since both interviewers and surveyors are native speakers of Spanish, the use of translators was not necessary

  1.  In the results section, it would be helpful to see the exact significance levels reported in either the main text or in the tables/figures.

Answer: The exact level of significance of each effect was included in the text. In the tables and figures, the following symbology was included *p < .05. **p < .01. ***p < .001, to identify the level of significance of each effect.

  1. In the results section on lines 255-257, there is an error in the reported results for the outcome positive relations because ethnic identity did not fully mediate the relationship between ethnic discrimination and positive relations since there was not a statistically significant direct effect between these variables.

Answer: Precisely because there is no statistically significant direct effect between ethnic discrimination and positive relationships (c'=0), the statistically significant indirect effect (a*b) exerted by ethnic identity is considered a complete or total mediation. On the contrary, in cases where there is a significant indirect effect and a significant direct effect, we speak of partial mediation.

  1. Similar for the outcome variable for autonomy, ethnic discrimination did not have a significant direct effect on this outcome variable, so ethnic identity cannot partially mediate a direct effect that did not occur.

Answer: As noted in Figure 2, ethnic discrimination exerted a statistically significant direct effect on autonomy (b = .13, p = .040), a relationship that in turn was mediated by the indirect effect of ethnic identity (b = -.03, p = .017, see Table 2).

  1. Given the cross-sectional nature of the data, the authors should avoid using causal language to describe the mediational results. In the discussion section, the authors describe their findings as occurring because ethnic discrimination caused people to feel more connected to their ethnic group, which in turn, lead to greater psychological outcomes.

Answer: The reviewer's suggestions are taken into account and it is drafted differently.

12.One question for the authors to further consider is whether the mediating processes they examined are dependent on the age of the participants. For example, prior research has shown that the developmental trajectories for racial/ethnic identities differ based on age and the developmental period of the participants’ life. As a result, if older adults are more stable in their racial/ethnic identity, they may be less likely to be susceptible to the mediating mechanisms described in this paper.

Answer: We find this observation extremely interesting, as it was not in our hypothesis we did not carry out the analysis, but as it seems important to us, we have included the author's idea in the discussion.

“…considering that previous research has shown that the developmental trajectories of racial/ethnic identities could differ according to the age and developmental period of the participants' lives, it would be interesting to evaluate in future research the possible moderating effect of age on the mediating effect of ethnic identity.

  1. In the introduction, the authors note that the abbreviation of ED and RD would be used to represent ethnic discrimination and racial discrimination, respectively. However, throughout the course of the manuscript the authors are inconsistent regarding their usage of ED, RD or the full phrases.

Answer: Thanks, we have corrected this across the text

  1. Likewise, there are other ways ED and RD are referenced during the paper that aren’t consistent. For example, in the description for Figure 1, but not figure 2, the authors say the relationship between “ethnic/racial discrimination”, whereas, in the paper the authors often refer to them as separate constructs (i.e., racial and ethnic discrimination). Given that these two variables are not collapsed into one, the authors should consistently separate these words out throughout the paper and figure descriptions.

Answer: Thanks, we have corrected this across the text

  1. Another instance where the authors deviate from their usual pattern of describing ED and RD is on line 241 where they phrase it as “ethnic-racial discrimination”. In addition, the authors suddenly begin to use the ED and RD abbreviations again on line 243 despite not using it throughout the majority of the paper. The authors often go back and forth between saying “ethnic and racial discrimination” and “racial and ethnic discrimination”

 Answer: Thanks, we have corrected the text

  1. Line 73: change “on color people” to “on people of color”

Answer: corrected

  1.  Line 198: Did the authors mean to write, “Sometimes ‘it’ prefers to hide that ‘it’ is Columbian”? Or should the “it” be replaced with “I” or “you”?

Answer: Thanks, we have corrected this across the text

In the case of the ethnic identity scale, although it did not present any heywood cases, it did show items with non-significant factor loads ('Sometimes he/she prefers to hide that he/she is Colombian')

  1. On line 259/260, I believe the authors meant to write that ethnic identity did not mediate the relationship between “racial discrimination and psychological well-being”, and not “ethnic discrimination”.

Answer: it is indeed a mistake, thank you very much, we have corrected the sentence.

  1.  There are several paragraphs that are only 2 or 3 sentences long and they impacted the overall flow of particular sections. I encourage the authors to find a better way to integrate these brief paragraphs for a smoother transition.  I recommend reviewing the paper for sentence structures, run-on sentences, oxford commas, and grammatical errors. In particular, there are a few grammatical errors in the paragraph starting on line 349.

Answer: We have corrected the wording and revised the text.